# Quality of Edible Sesame Oil as Obtained by Green Solvents: In Silico versus Experimental Screening Approaches

**DOI:** 10.3390/foods12173263

**Published:** 2023-08-30

**Authors:** Sinda Trad, Emna Chaabani, Wissem Aidi Wannes, Sarra Dakhlaoui, Salma Nait Mohamed, Saber Khammessi, Majdi Hammami, Soumaya Bourgou, Moufida Saidani Tounsi, Anne-Sylvie Fabiano-Tixier, Iness Bettaieb Rebey

**Affiliations:** 1Laboratory of Aromatic and Medicinal Plants, Borj Cedria Biotechnology Center, BP. 901, Hammam-Lif 2050, Tunisia; sinda.trad@etudiant-fst.utm.tn (S.T.); aidiwanneswissem18@gmail.com (W.A.W.); saradakhlaoui@gmail.com (S.D.);; 2GREEN Extraction Team, Université d’Avignon et des Pays de Vaucluse, INRA, UMR408, 84000 Avignon, France; 3Laboratory of Olive Biotechnology, Borj Cedria Biotechnology Center, Hammam-Lif 2050, Tunisia; salma.nait@cbbc.rnrt.tn

**Keywords:** *Sesamum indicum* L., solvent extractions, seed oils, fatty acids, sterols, tocopherols, phenolics, antioxidant potential, anti-inflammatory activity

## Abstract

The present study aimed to investigate the qualitative and quantitative performance of five green solvents, namely 2-methyltetrahydrofuran (MeTHF), cyclopentyl methyl ether (CPME), *p*-cymene, *d*-limonene and ethanol to substitute *n*-hexane, for sesame seed oil extraction. In fact, both CPME and MeTHF gave higher crude yields than *n*-hexane (58.82, 54.91 and 50.84%, respectively). The fatty acid profile of the sesame seed oils remained constant across all the solvent systems, with a predominance of oleic acid (39.27–44.35%) and linoleic acid (38.88–43.99%). The total sterols gained the upmost amount with CPME (785 mg/100 g oil) and MeTHF (641 mg/100 g oil). CPME and MeTHF were also characterized by the optimum content of tocopherols (52.3 and 50.6 mg/100 g oil, respectively). The highest contents of total phenols in the sesame seed oils were extracted by CPME (23.51 mg GAE/g) and MeTHF (22.53 mg GAE/g) as compared to the other solvents, especially *n*-hexane (8 mg GAE/g). Additionally, sesame seed oils extracted by MeTHF and CPME also had the highest antioxidant and anti-inflammatory properties as compared to the other green solvents and *n*-hexane, encouraging their manufacturing use for sesame seed oil extraction.

## 1. Introduction

Sesame (*Sesamum indicum* L.) seeds are among the oldest oilseed crops, playing a crucial role in human nutrition [1]. Edible sesame seed oil has exceptional nutritional value due to its richness in essential fatty acids as linoleic acid. In addition, sesame is a rich source of protein and several health-promoting compounds, such as phytosterols, tocopherols, and lignans [2]. This peculiar biochemical composition makes sesame oil one of the most resistant vegetable oils against oxidation [3]. 

Notably, the potential health benefits of consuming sesame or sesame products were experimentally approved for their anti-inflammatory and anti-carcinogenic effects, as well as for their positive impacts on blood pressure and serum lipid levels and their inhibition of in vitro human colon cancer cell proliferation [4,5]. Approximately 70% of the world’s sesame seed production is specifically dedicated to sesame oil production [1]. Indeed, its proportion of the consumption of total annual oil and food is about 65% and 35%, respectively [6]. Vegetable oils are crucial for food formulation, adding flavor and texture to food. Moreover, these oils are important nutrients for human health, being a source of energy and essential fatty acids (linoleic acid and linolenic acid) [7]. 

In addition, the quality of vegetable oils is related to the presence of bioactive compounds, the contents of which may vary according to the extraction process. In general, the extraction method that uses solvents is one of the most widely used in the industry, due to high oil yield. Conventionally, *n*-hexane has been used as the best available solvent over the last five decades for oilseed extraction, owing to its chemical stability, high selectivity to oils, limited energy cost, low boiling point and polarity [8]. However, increasing concerns regarding *n*-hexane toxicity, environmental pollution, and its non-renewable nature have led the European Union to impose stricter regulations. The compound *n*-hexane has been classified as a neurotoxin and suspected reprotoxic substance under the REACH regulation [9,10]. 

To meet the requirements of regulations in the field of food supplements, we have to use safe solvents for the extraction of ingredients/extracts destined for human consumption/nutrition, such as green solvents. The green extraction concept originated from green chemistry [11]. Thus, solvents alternative to petrochemical solvents have currently become an emerging trend for both academics and industries in terms of safety and environmental and economic considerations. Nonetheless, long-term exposure to *n*-hexane and its leakage during industrial extraction and recovery may cause negative human health and environmental problems [12,13,14]. 

Computational methods such as Hansen solubility parameters (HSP) [15] and the COnductor-like Screening MOdel for Real Solvents (COSMO-RS) [16] can be used for the salvation power investigation. The use of green solvents for lipid extraction from plants has been successfully evaluated [17,18,19,20,21]. Although there have been numerous studies published on sesame seeds, the investigation of the use of green solvents for sesame seed oil extraction is still limited. Only one study was found as of today that used enzymes as an effective and safe green extraction method for sesame seed oil extraction [22].

There is no doubt that our paper aimed for the first time to investigate, through a computational approach, the effect of using alternative solvents (MeTHF, CPME, *p*-cymene, *d*-limonene and EtOH) instead of *n*-hexane for sesame seed oil extraction. Subsequently, the extracted oils were compared to those obtained with a conventional solvent (*n*-hexane) in terms of oil yields, physicochemical properties, and antioxidant and anti-inflammatory activities. 

## 2. Materials and Methods

### 2.1. Reagents and Plant Material

The following solvents were obtained from Sigma Aldrich (Sigma Aldrich, Steinheim, Germany): 2-Methyltetrahydrofuran (≥99%), Cyclopentyl methyl ether (99.9%), p-Cymene (99%), d-Limonene (97%), ethanol (≥99.8%), Methanol, isopropanol, hexane, cyclo-hexane, acetonitrile, and methyl tert-butyl ether (suitable for HPLC, ≥99%). The reagents Potassium hydroxide (ACS reagent, ≥85%, pellets), Acetic acid (glacial, ACS reagent, ≥99.7%), Folin–Ciocalteu’s phenol reagent (suitable for the determination of total protein by the Lowry method, 2 N), Gallic acid (97.5–102.5% for titration), and 1,1-diphenyl-2-picrylhydrazyl (97%) were also acquired from Sigma Aldrich (Sigma Aldrich, Steinheim, Germany).

Sesame (*Sesamum indicum* L.) seeds were harvested in June 2022 from Korba region in the northeastern part of Tunisia (latitude 36°34′38.22″ (N); longitude 10°51′29.63″ (E); altitude 637 m). 

### 2.2. Computational Methods

#### 2.2.1. Solute–Solvent Solubility Prediction by HSPs 

The solubility of major metabolites typically found in sesame seed oil were predicted in *n*-hexane and five green solvents (MeTHF, CPME, *p*-cymene, *d*-limonene and EtOH) using the theory of Hansen solubility parameters (HSPs). This computational approach, developed by Hansen [15], has been widely utilized as a practical decision-making tool based on the total (Hildebrand) solubility parameters to explain dissolution behavior [21,23,24,25]. It provides an efficient and convenient way to describe solvent–solute interactions based on the principle of “like dissolves like” [15]. 

According to the Hansen model, the overall cohesive energy density is governed by the cumulative energies needed to counter dispersion forces ($\delta_{d^{2}}$), polar forces resulting from dipole moments ($\delta_{p^{2}}$), and interactions involving hydrogen bonding (electron exchange, proton donor/acceptor) among molecules ($\delta_{h^{2}}$), as described by Equation (1).
(1)$$\delta_{{\mathrm{total}}^{2}}=\delta_{d^{2}}+\delta_{p^{2}}+\delta_{h^{2}}$$

Here, δ_total_ represents the Hansen total solubility parameter, which consists of three HSPs: δ_d_ (dispersive term), δ_p_ (polar term), and δ_h_ (hydrogen bonding term). To optimize HSP solvents, a composite affinity parameter called the relative energy difference number (RED) was calculated. This parameter determines the solubility between the solvent and the solute, and it is expressed as below (Equation (2)):RED = R_solv_/R_spher_(2)

R_spher_ represents the radius of the Hansen solubility sphere, while R_solv_ corresponds to the distance of a solvent from the center of the Hansen solubility sphere, calculated using Equation (3): (3)$$R_{\mathrm{solv}}={\left(4{\left(\delta_{\mathrm{dSolu}}-\delta_{\mathrm{dSolv}}\right)}^{2}+{\left(\delta_{\mathrm{pSolu}}-\delta_{\mathrm{pSolv}}\right)}^{2}+{\left(\delta_{\mathrm{hSolu}}-\delta_{\mathrm{hSolv}}\right)}^{2}\right)}^{1/2}$$

In this equation, “Solu” refers to the solute, and “Solv” refers to the solvent.

Generally, this parameter follows the classical principle of “like dissolves like”, where a smaller R_solv_ value indicates a higher expected affinity between the solute and the solvent. Therefore, a suitable solvent has a RED number less than 1, exhibiting favorable properties for dissolution, while an unsuitable solvent has a RED number greater than 1.

The JChemPaint version 3.3 software (GitHub Pages, San Francisco, CA, USA) was used to convert the chemical structures of the solvents and solutes into simplified molecular input line entry syntax (SMILES) notations. These SMILES notations were then employed in the HSP calculation using the Yamamoto-molecular break method, which breaks down the SMILES into corresponding functional groups and estimates their Hansen solubility parameters (HSPs). This method is embedded in the HSPiP software 5th edition 5.3.07 to facilitate the calculation of HSPs for target metabolites and tested solvents. 

#### 2.2.2. COSMO-RS Prediction 

The COnductor-like Screening MOdel for Realistic Solvents (COSMO-RS), developed by Klamt [16], is a computational approach that combines quantum chemical calculations (COSMO) with statistical thermodynamics (RS). This method allows for the determination and prediction of the chemical potential of the molecules in a liquid phase, eliminating the need for experimental data. Previously, this model has been applied in the extraction field to assess the chemical potential of hydrophobic molecules in green solvents [14,19,20,26]. By implementing the COSMO-RS model in COSMOtherm software (C30 1401, CosmothermX14, COSMOlogic GmbH & Co. KG, Leverkusen, Germany), the relative solubility of target metabolites (palmitic acid (C16:0), oleic acid (C18:1), linoleic acid (C18:2), γ-tocopherol and β-sitosterol) in tested solvents was calculated.

The calculation of relative solubility is derived from the following equation:(4)$$\log_{10}{(x}_{j})=\log_{10}\left[\frac{\exp\left({µ}_{j}^{pure}-{µ}_{j}^{solvent}-\Delta G_{j,fusion}\right)}{RT}\right]$$

$\mu_{j}^{pure}:chemical\;potential\;of\;pure\;compound\;j\;(Joule/mol)$;

$\mu_{j}^{solvent}:chemical\;potential\;of\;j\;at\;infinite\;dilution\;(Joule/mol)$;

$\Delta G_{j,fusion}:free\;energy\;of\;fusion\;of\;j\;(Joule/mol)$;



$x_{j}:solubility\;of\;j\;(g/g\;solvent).$



The relative solubility is determined at infinite dilution. The logarithm of the best solubility is set to 0, and all other solvents are ranked relative to the best or reference solvent.

### 2.3. Oil Extraction

Sesame seed oils were extracted under reflux for 8 h with the selected solvents (*n*-hexane, MeTHF, CPME, ethanol, limonene and *p*-cymene). These solvents were selected for their hydrophobicity, technical properties and previous effectiveness on the alternative solvents. These could later be suitable for lipid extraction from sesame seeds and could be good candidates to replace *n*-hexane. Three replicates of the experiments were carried out.

The corresponding extraction yield was expressed as the percentage of the mass of crude oil obtained relative to the mass of dry sesame seeds used for extraction.
$$Extraction\;yield\;\left(g/100\;g\;DW\right)=\frac{Mass\;of\;crude\;oil\;(g)}{Mass\;of\;dry\;seeds\;(g)}\times 100$$

### 2.4. Physicochemical Characteristics of Sesame Seed Oils

Acid value, refractive index, peroxide value and iodine value of sesame seed oils were measured based on official methods of AOCS [27].

#### 2.4.1. K232 and K270 Determination

Extinction coefficients (K232 and K270) were determined according to (AOCS Ch 5-91) [28]. 

#### 2.4.2. Determination of Chlorophyll Content

The chlorophyll concentrations in the oil samples were assessed in cyclo-hexane following the protocol of Chtourou et al. [29]. 

#### 2.4.3. Oxidative Stability

The method of Tabee et al. [30] was used to determine the oxidative stability of sesame seed oils.

### 2.5. Fatty Acid Composition

Fatty acid composition was analyzed by gas chromatography (GC) after derivatization to fatty acid methyl esters (FAMEs) with a 2 M methanolic solution of potassium hydroxide [18]. 

### 2.6. Sterol Analysis 

The contents and composition of sterols were determined by GC following the procedure described by Bettaieb Rebey et al. [18]. 

### 2.7. Tocopherol Contents 

Tocopherols content was determined according to the Bourgou et al. [17] procedure. 

### 2.8. Total Phenolic Content 

The quantification of total phenol content (TPC) in the oil samples was carried out using the Folin–Ciocalteu reagent method. Initially, methanol was employed to extract the total polyphenols from the oil samples [31]. The measurement of total polyphenol content is presented as milligrams of gallic acid equivalent per gram of oil (mg GAE/g). Each sample underwent three replicate tests for accuracy.

### 2.9. Antioxidant Potential Analyses

The total antioxidant capacity (TAC) of the oil extracts was evaluated through the assay of the green phosphate/Mo^5+^ complex [32]. The absorbance was measured at 695 nm against a blank. The total antioxidant activity was expressed as mg GAE/g oil.

The potential of methanolic oil extracts to reduce the free DPPH radical (1,1-diphenyl-2-picrylhydrazyl) was expressed as IC_50_ (µg/mL)—the anti-radical dose required to cause a 50% inhibition [31]. The ferric reducing antioxidant power (FRAP) of oil extracts was determined following the method described by Benzie and Strain [33].

### 2.10. Anti-Inflammatory Activity 

#### 2.10.1. Cell Culture 

RAW 264.7 murine macrophage cells were supplied from the American Type Culture Collection (ATCC, Manassas, VA, USA). The cell line was cultured in RPMI 1640 medium (Gibco Bio-cult, Glasgow, UK) supplemented with 10% (*v*/*v*) fetal bovine serum (Gibco, Carlsbad, CA, USA), 100 U/mL of penicillin, 100 mg/mL of streptomycin (Sigma-Aldrich, Deisenhofen, Germany). 

#### 2.10.2. Measurement of Nitrite Production 

RAW 264.7 cells were seeded in 24-well plates at a density of 2105 cells per well and were allowed to attach for 24 h at 37 °C. Then, after 60 min, treated cells with increasing concentrations (100, 200, 400 µg/mL) of extracts (hexane, MeTHF, CPME), dissolved in the DMSO were stimulated with 100 mg/mL lipopolysaccharide (LPS). Cell controls were stimulated by LPS without extract treatment. 

After 24 h, the quantity of nitrite accumulated in the culture supernatant was determined based on the Griess reaction [34]. NO production by controls and cells treated with sesame oils were compared.

### 2.11. Statistical Analysis

The results obtained from the analytical methods were presented as the mean ± standard deviation (SD). A Duncan test (*p* < 0.05) was conducted to identify significant variations among the means of the experiments, which were conducted in triplicate. For the cell-based experiments, the data were expressed as the standard error of the mean (SEM), and the values were obtained from at least three separate experiments. A statistical analysis of variance (ANOVA), followed by a Bonferroni’s test, was utilized for multiple comparisons of the data. Statistical significance was attributed to *p*-values lower than 0.01.

## 3. Results and Discussion

### 3.1. Computational Methods

In this study, the solubility of the major hydrophobic components of sesame seed oil, including palmitic acid, oleic acid, linoleic acid, γ-tocopherol, and β-sitosterol, was predicted using two decision-making tools: the Hansen solubility parameters and COSMO-RS. The solubility calculations were performed in various tested solvents, and the properties of these solvents were presented in Table 1. The solvents included in the study were *n*-hexane (serving as the reference solvent), two ethers (MeTHF and CPME), two terpenes (*p*-cymene and *d*-limonene) and an alcohol (ethanol).

The relative energy difference (RED) values estimated the capacity of selected solvents to dissolve the major lipid components (C16:0, C18:1, C18:2, γ-tocopherol and β-sitosterol) of sesame seed oil and are recapitulated in Table 2. 

Among these green solvents, *d*-limonene was theoretically the most suitable solvent to replace *n*-hexane, followed by MeTHF, CPME and *p*-cymene. In contrast, EtOH was worse than *n*-hexane (RED > 2), and it seems to be worse as a substitute for *n*-hexane for the extraction of these metabolites, as observed by Cascant et al. [14] for lipid extraction from salmon fish.

### 3.2. COSMO-RS Simulations

A COSMO-RS simulation was additionally conducted to determine the relative solubility, log10 (Xj), of the major components of sesame seed oil (traditionally extracted with *n*-hexane) in the tested solvents. The results of the COSMO-RS simulations were summarized in Table 3. 

It is evident that all bio-based solvents exhibited a strong potential for replacing *n*-hexane for the extraction of fatty acids (C16:0, C18:1, C18:2), γ-tocopherols (except EtOH, which performed worse than *n*-hexane), and β-sitosterol. Notably, the green solvents demonstrated higher solubilization power than *n*-hexane, as indicated by their higher log_10_ (Xj) values. A similar trend was observed by Chaabani et al. [19] for the extraction of lentisk fruit oil, by Cascant et al. [14] for the extraction of salmon fish oil, and by Sicaire et al. [20] for the extraction of rapeseed oil. Considering all constituents and candidate solvents, MeTHF and CPME appeared to be the most suitable alternative solvents, as their log10 (Xj) values were null.

According to the in silico screening using HSP parameters and COSMO-RS predictions, among all the studied solvents, MeTHF, CPME and *d*-limonene emerged as the most promising alternatives to conventional *n*-hexane for extracting the major metabolites of sesame seed oil. These findings were further compared with experimental results to establish correlations between the actual extraction of sesame seed oil and the computed results using HSP parameters and COSMO-RS.

### 3.3. Physicochemical Properties

The physicochemical properties of sesame seed oils extracted by *n*-hexane and five agro-solvents are given in Table 4. The type and polarity of the solvents significantly influenced the different physicochemical properties of the sesame seed oils. 

The acid value of the sesame seed oil extracted by CPME was the highest (1.73 mg KOH/g), while the oil samples extracted by *d*-limonene revealed the lowest (0.92 mg KOH/g). Higher acid values were obtained by Olaleye et al. [35] in the case of Nigerian sesame seed oils extracted by cold press (6.09 mg KOH/g) and an *n*-hexane solvent (5.89 mg KOH/g). Similarly, Paul [36] found that the acid value of Nigerian sesame seed oil extracted by cold press was 5.46 mg KOH/g. Acid value mainly reflects the refining degree of oil and the preservation of raw materials. In addition to free fats, pigments, and phospholipids, other substances may be extracted during the extraction of sesame seeds. Meanwhile, increasing the solvent polarity accelerated the destruction of lipids and free fatty acids binding to lipoproteins or cell membranes [37]. Hence, the levels of free fatty acids and fatty alcohols in oils extracted by stronger polar solvents might be higher, resulting in higher acid values. It is also interesting to mention that the acid value is a reflection of the pH value of oil; that is, as the acid value increases, the pH of the oil decreases [36]. In fact, the acid value of a strong polar solvent is generally higher than that of a weak polar solvent, but in our study, the polar *n*-hexane solvent showed the second highest value (1.69).

As shown in Table 4, the peroxide value of the sesame seed oil extracted by MeTHF was the lowest (2.67 mEq O_2_/kg), and with *d*-limonene, it was the highest (4.83 mEq O_2_/kg). Higher peroxide values were observed from cold press (5.84 meq O_2_/kg)- and *n*-hexane solvent (5.61 meqO_2_/kg)-extracted Nigerian sesame oils [35]. The peroxide values of cold-pressed oils from Pakistan white, black and brown sesame seeds were 4.46, 5.51 and 9.12 mEq O_2_/kg, respectively [38]. The obtained peroxide value of *n*-hexane-extracted sesame oil (3.42 mEq O_2_/kg) was higher than that obtained by Paul [39] in the case of Nigerian sesame oil extracted by *n*-hexane (2 mEq O_2_/kg). 

Contrary to the peroxide results, the iodine value of the sesame seed oil extracted by MeTHF was the highest (123 g/100 g), and with *d*-limonene, it was the lowest (93 g/100 g), as given in Table 4. These values were comparable to the iodine values of some local Sudanese and imported sesame seed cultivars, which varied from 101.52 to 114.85 g/100 g for the local cultivars, and 97.70 to 111.30 g/100 g for the introduced cultivars, as reported by El Khier et al. [39]. Olaleye et al. [35] mentioned that significant differences (*p* < 0.05) in iodine values existed between cold press (83.73 g I_2_/100 g)- and *n*-hexane solvent (92.38 g I_2_/100 g)-extracted Nigerian sesame seed oils. Nzikou et al. [40] found that the iodine value of Northern Congo sesame seed oil extracted by petroleum ether solvent was 117.2 g I_2_/100 g. Algerian sesame seed oil extracted by petroleum ether solvent had an iodine value equal to 113.11 g I_2_/100 g [2]. In general, oil has a certain iodine value based on the number of unsaturated double bonds in it, and the greater the number, the higher the value. In some vegetable oils, iodine values range from 90 to 130, for example, soybean oil, corn oil, and canola oil [41].

In Table 4, the differences in oxidative stability, the refractive index, and the K232, K270 and chlorophyll values of sesame seed oils extracted by *n*-hexane and five green solvents were not significant (*p* > 0.05). Also, for K270, no significant differences were observed between solvents, with the exception of *n*-hexane and ethanol. Similar results were obtained by Olaleye et al. [35], who found that there was no significant difference (*p* > 0.05) observed between the refractive index of cold press (1.469)- and *n*-hexane solvent-extracted Nigerian sesame seed oils (1.468). The refractive index of cold press sesame and *n*-hexane solvent-extracted Tunisian sesame seed oils was slightly lower (1.471), as reported by Elleuch et al. [42]. The refractive index, K232 and K268 of Algerian sesame seed oil extracted by petroleum ether were 1.476, 3.12 and 0.65, respectively [2]. According to Paul [39], the refractive index is mainly used to measure the change in unsaturation as the fat or oil is hydrogenated. The refractive index of oils depends on their molecular weight, fatty acids chain length, degree of unsaturation and degree of conjugation. 

It could be deduced that the physicochemical properties of the sesame seed oils were recognized to depend on the polarity of the solvent extraction and on additional factors, such as plant cultivar, environmental fluctuations, and growing site.

### 3.4. Experimental Solvent Screening

#### 3.4.1. Oil Yield 

After quantitatively extracting sesame seeds for 8 h with *n*-hexane, MeTHF, CPME, Ethanol, limonene and *p*-cymene, extraction yields were determined gravimetrically. As can be seen in Figure 1, the extraction yields were within the 24.64–58.82% range, which was in accordance with the average lipid content of sesame seeds, around 45–60% [10,43]. 

Interestingly, the economic value of sesame seeds is dependent on their high oil content. Indeed, the assessment of substituting the toxic *n*-hexane, conventionally used for lipid extraction, with alternative solvents derived from renewable feedstocks has been conducted, revealing significant differences between MeTHF, CPME, ethanol, d-limonene, and *p*-cymene. Thus, both CPME and MeTHF gave higher crude yields than *n*-hexane (58.82, 54.91 and 50.84%, respectively). Furthermore, ethanol and *p*-cymene resulted in the lowest lipid yields (27.29 and 24.64%, respectively). These variations in extraction yields could be attributed to the difference between the abundance of polar and non-polar compounds in sesame seeds.

Fatty acid composition is an essential indicator of the nutritional value of the oil. The extracted seed oils were analyzed by GC-FID, and the proportions of fatty acids were determined (Figure 2). 

Table 4 shows that the fatty acid profile of sesame seed oils remained constant across all the solvent systems tested, indicating that choice of solvent was not specific for any certain fatty acid. Therefore, it can be seen that the predominant fatty acids were oleic (C18:1) (39.27–44.35%) and linoleic (C18:2) (38.88–43.99%), which represent more than 82% of the total fatty acids in extracted oil. In addition, there are some fatty acids with low proportions, such as palmitic acid (C16:0), stearic acid (C18:3), linolenic acid (C18:0), palmitoleic acid (C16:1), eicosenoic acid (C20:1). Thus, sesame seed oil belongs to the oleic–linoleic acid group. The fatty acid composition evaluated in Tunisian sesame seeds was very similar to previously reported ranges [2,10] and was found to be satisfactory in terms of the official Standard of Codex Alimentarius [44].

The total saturated fatty acids (SFAs) of sesame seed oils varied between 12.59 and 18.10%, whereas the monounsaturated fatty acids (MUFAs) ranged from 39.82 to 44.76%, and the PUFAs ranged from 39.52 to 44.32%. Indeed, ether oil extracts (MeTHF and CPME) were the best alternative to *n*-hexane for the extraction of sesame seed oil. The relatively low percentage of (C18:2) obtained with *d*-limonene can be explained by the amount of solvent left after the evaporation step [20]. The PUFAs are considered healthier than the saturated ones [45]. Hence, the evaluation of the PUFA/SFA ratios shows that CPME proved to be the best agro-solvent for the extraction of sesame oil with high nutritional quality.

#### 3.4.2. Sterol Composition 

The determination of sterol composition is of major interest due to their antioxidant activity and their impact on health [46]. 

The total sterol content of sesame oil ranged from 543–785 mg/100 g (Table 5), which was found to be in the range of previously published values for sesame seed oil from other countries reported in the literature (408.9–620.5 mg/100 g) [10]. 

In fact, the contribution of total sterols reached the upmost amount with CPME (785 mg/100 g oil) and MeTHF (641 mg/100 g oil), in comparison to the reference (Table 5). 

Among the sterols, β-sitosterol and campesterol were found to be the major components of sesame seed oil. Moreover, the minor sterols were D5-avenasterol, stigmasterol, D7-avenasterol and cholesterol. Regarding the relative contents of sterol distribution in the oil extracted with green solvents and the reference, CPME exhibited the highest β-sitosterol content (555 mg/100 g), followed by MeTHF (437 mg/100 g). In addition, the content of campesterol, the next major component, was enhanced 1.6- and 1.25-fold compared to that obtained by the conventional solvent. Overall, phytosterol content in Tunisian sesame seeds, independent of the solvent used, was found to be satisfactory and within the range of the published values of sesame oils [10,39]. 

Β-sitosterol is recognized for its positive impact on human health and physiological well-being. It has been attributed to the ability to reduce cholesterol levels, enhance immune function, and possess anti-inflammatory, antipyretic, and anti-carcinogenic properties [47].

#### 3.4.3. Tocopherol Composition 

Numerous research studies have highlighted a connection between the characteristics of vegetable oils and the levels of tocopherols. As a result, the ability to accurately measure the concentrations of these natural antioxidants holds significant importance [48].

Total tocopherol content in sesame seed oil ranged from 39.8 to 52.3 mg/100 g. Only α-, γ- and δ-tocopherols were present in the sesame seed oil (Table 5). Γ-tocopherol was the main component, followed by δ-tocopherol and α-tocopherol. 

These results are in agreement with the findings of Gharby et al. [10] for Morrocan sesame seeds. It is worth mentioning that among the green solvent used in this work, CPME and MeTHF could extract more tocopherols than *n*-hexane (52.3 and 50.6 mg/100 g, respectively) (Table 5). Hence, the amount of the major isomer γ-tocopherol was improved by 15 and 10%, respectively, for the CPME and MeTHF oil extracts. However, the effect of bio-based solvents on tocopherol content has been reported by several studies. Interestingly, tocopherols in kernel oil extracted by green solvents (DMC (dimethyl carbonate); CPME) were found to be much higher than those extracted by *n*-hexane [49]. Additionally, Bourgou et al. [17] reported a high amount of tocopherols by means of MeTHF in comparison with a conventional solvent for basil and black cumin seeds. According to Sicaire et al. [20], the contents of the tocopherols in rapeseed oil were similar with MeTHF and *n*-hexane extraction.

In summary, sesame oils obtained by CPME and MeTHF exhibited the highest tocopherol content in comparison to the conventional solvent. This could be explained by the difference in polarity and chemical structures of the main tocopherols present in sesame seed oil.

#### 3.4.4. Total Phenolic Content

Polyphenols are well known for their powerful antioxidant properties and ability to inhibit lipid peroxidation. In Table 6, the amount of total phenols varied among sesame seed oils using different solvents. In view of the results, it was clear that the total amount of phenols varied significantly depending on the nature of the extraction solvent used (*p* < 0.05). The highest phenol totals were extracted by CPME, MeTHF, *d*-limonene and ethanol (23.51, 22.53, 22.35, 21.54 mg GAE/g, respectively), while the lowest were by *p*-cymene (11.35 mg GAE/g) and *n*-hexane (8 mg GAE/g).

These results suggested that green procedures, namely CPME, MeTHF, *d*-limonene and ethanol, had better selectivity and yielded oil with a higher amount of total phenols than did *n*-hexane solvent. Bopitiya and Madhujith [50] found that the total phenol content of Sri Lanka sesame seed oil extracted by a methanol solvent was 26 mg GAE/g. However, Algerian sesame seed oil extracted by a petroleum ether solvent had a total phenol content equal to 152.2 mg GAE/g [2]. The total phenol content of 32 cold-pressed sesame seed oils manufactured in Poland varied between 84.03 and 103.79 mg GAE/g [51]. The total phenol content of Nigerian white sesame seed oil extracted by a *n*-hexane solvent was 196.44 mg GAE/g [52].

#### 3.4.5. Antioxidant Potential

As shown in Table 6, the highest total antioxidant potential of the sesame seed oils was obtained by CPME (3.92 mg GAE/g) and MeTHF (3.14 mg GAE/g) as compared to the other solvents (1.92–2.05 mg GAE/g). The total antioxidant potential of the sesame seed oils was significantly correlated with total sterol (*r* = 0.959), total tocopherol (*r* = 0.841) and total phenol (*r* = 0.600) contents. Sesame seed oils extracted using CPME (IC_50_ = 30.71 μg/mL) and MeTHF (IC_50_ = 90.02 μg/mL) showed stronger radical scavenging potential than the other green and *n*-hexane solvents with IC_50_ ˃ 100 μg/mL. The anti-radical activity of the sesame seed oils was significantly correlated with total sterol (*r* = −0.913), total tocopherol (*r* = 0.700) and total phenol (*r* = −0.666) contents. Additionally, sesame seed oils extracted using CPME (IC_50_ = 435.93 μg/mL), *d*-limonene (IC_50_ = 580.62 μg/mL) and MeTHF (IC_50_ = 670.15 μg/mL) showed stronger reducing power potential than the other green and *n*-hexane solvents with IC_50_ ˃ 800 μg/mL. The reducing power activity of the sesame seed oils was significantly correlated with total sterol (*r* = −0.882), total phenol (*r* = −0.739) and total tocopherol (*r* = −0.600) contents. So, these results suggested that the antioxidant potential of sesame seed oils could be due to the presence of bioactive substances such as phenols, tocopherols and sterols.

Arab et al. [2] found that the total antioxidant potential of Algerian sesame seed oil extracted by petroleum ether (1.39 mg GAE/g) was comparable to that found in our study in the case of *n*-hexane (1.94 mg GAE/g). Bopitiya and Madhujith [50] reported that methanolic sesame seed oil from Sri Lanka possessed a high quantity of phenolics, and its antioxidant potential as measured by DPPH assay was significantly higher (IC_50_ = 0.026 mg/mL) than that of the positive control α-tocopherol (IC_50_ = 0.030 mg/mL).

### 3.5. In Silico Screening versus Experimental Screening

Computational simulations were compared with the experimental results in order to establish a correlation between the in silico approach and the experimental approach and to evaluate the potential of green solvents to replace the toxic *n*-hexane conventionally used in lipid extraction.

The HSP and COSMO-RS calculations showed that the most suitable solvents for *n*-hexane substitution were MeTHF, CPME and *d*-limonene. The experiments indicated that among these potential candidates, only CPME and MeTHF are promising alternatives to *n*-hexane replacement. According to the HSP and COSMO-RS calculations, MeTHF, CPME, and d-limonene emerged as the most suitable solvents for replacing conventional solvents. Among these three potential candidates, CPME and MeTHF stood out as promising alternatives, considering overall global yield and quality of oil. Thus, both CPME and MeTHF yielded higher crude oil than *n*-hexane (58.82% and 54.91%, respectively), whereas limonene resulted in a lower lipid yield (37.89%).

The utilization of *d*-limonene in oil extraction presented certain drawbacks linked to the presence of residual *d*-limonene in the sample and the higher temperature needed to reach its boiling point (175.4 °C) [53].

Economically and technically, CPME and MeTHF had properties similar to those of *n*-hexane and, more specifically, for the energy required to evaporate 1 kg of solvent and its carbon footprint. Furthermore, CPME is an eco-friendly solvent that has better stability to peroxide formation, low volatility, low water solubility, and acidic stability. It has been demonstrated in many applications, including furfural synthesis, secondary thioamide synthesis, carotenoid extraction, peptide synthesis, protein purification, and radical additions [54].

In the same way, MeTHF is a green solvent, is derived from renewable resources, has low toxicity [55] and has been approved for use in food industries by Scott et al. [56]. Against this background, the potential for CPME and MeTHF as alternative solvents for the extraction of vegetable oils has been reported in many studies. In this context, Probst et al. [57] demonstrated that CPME could be used as a safer solvent to substitute *n*-hexane for lipid extraction from the wet biomass of oleaginous yeast. As well, CPME and MeTHF appeared to be good candidates to replace *n*-hexane for the solubilization of microbial oils [25]. Likewise, MeTHF proved to be a good solvent to replace *n*-hexane for oil extraction from date nuclei, anise, fennel, carvi, black cumin and basilic seeds [17,18,21].

### 3.6. Anti-Inflammatory Activity

Based on the previous results, the anti-inflammatory activity of only CPME, MeTHF and *n*-hexane (taken as reference) was evaluated. The cytotoxic effects of conventionally and green-extracted oils on RAW264.7 cells were assessed, and the cells were treated with different sample concentrations (100–400 µg/mL). The results (Figure 3) showed that for the different solvents tested, concentrations of 100, 200 and 400 µg/mL had a negligible effect on RAW cells. Indeed, cell viability was higher than 80%. Therefore, these concentrations of *n*-hexane-, MeTHF- and CPME-obtained oils were applied for the subsequent experiment.

Non-cytotoxic conventionally and green-extracted oils were examined to study their potential to inhibit NO production in LPS-treated RAW 264.7 macrophages. LPS can activate macrophage cells to initiate inflammation and the production of pro-inflammatory mediators such as NO. The oils showed variable and moderate anti-inflammatory activity, depending on tested concentration (Figure 4). For the three oils, the high concentration of 400 µg/mL gives the highest activity. Conventionally extracted oil by *n*-hexane inhibited NO production by 37%. However, green-extracted oils using MeTHF and CPME exhibited higher anti-inflammatory activity, inhibiting NO production in LPS-stimulated macrophages by 43 and 44%, respectively. Theses oils also displayed higher activity than *n*-hexane-extracted oil at a dose of 200 µg/mL with inhibition percentages of 32 and 37%, respectively.

Our findings align with previous research that demonstrated the potent anti-inflammatory effects of sesame oil. This was evident through the suppression of inflammatory markers in scenarios like LPS-induced inflammation in RAW macrophages or mouse peritoneal macrophages. The observed anti-inflammatory properties of sesame oil were linked to the attenuation of pro-inflammatory cytokines [58,59]. However, the superior activity of green-extracted oils using MeTHF and CPME compared to conventionally extracted oil could be due to the presence of more active compounds compared to *n*-hexane-extracted oil. Our results showed that the green extracted oils displayed a higher phytosterol content than *n*-hexane-extracted oil (Table 5). Phytosterols were reported to exhibit anti-inflammatory effects and can reduce the expression of pro-inflammatory mediators such as NO and interleukin-1β (IL-1β), IL-6 and TNF-α in LPS-stimulated RAW264.7 macrophages [60,61]. On the other hand, the sesame oils extracted in our study were rich in both polyunsaturated fatty acids and monounsaturated fatty acids. Linolenic acid has an inhibitory effect on the production of NO [62]. It inhibits inducible nitric oxide synthase, COX-2, and TNF-α gene expressions induced by LPS [63].

## 4. Conclusions

Through a combination of experimental and theoretical validation, this study conclusively demonstrated that CPME and MeTHF serve as viable and suitable alternative solvents for the extraction of sesame seed oil, effectively replacing the traditionally employed toxic *n*-hexane. The utilization of CPME and MeTHF as solvents yielded noticeable improvements in both the quality and quantity of sesame seed oils, attributed to their heightened content of bioactive compounds, functional attributes, and biological properties. This transition towards CPME and MeTHF holds the potential to drive further innovations and their integration into the realm of food manufacturing. However, it is crucial for subsequent research to delve into the exploration of storage effects on the quality of sesame seed oil, specifically by investigating the oxidative stability of oils extracted using CPME and MeTHF.

## Figures and Tables

**Figure 1 foods-12-03263-f001:**
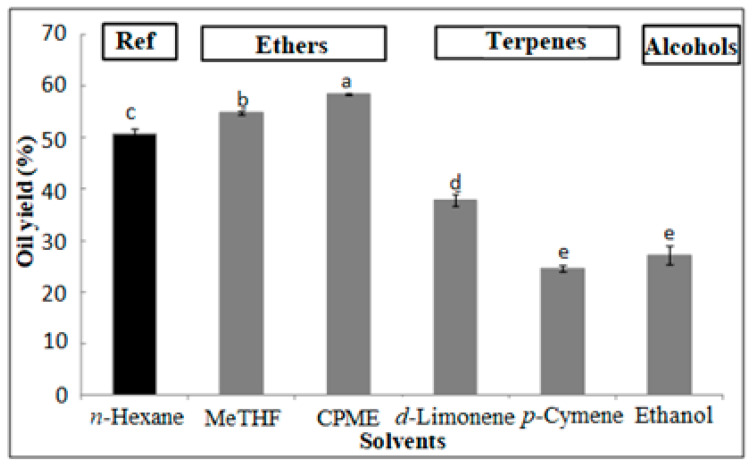
Sesame oil yields as obtained by eco-friendly solvents and compared to the reference. a–e letters = significant differences at *p* < 0.05. Fatty acid composition.

**Figure 2 foods-12-03263-f002:**
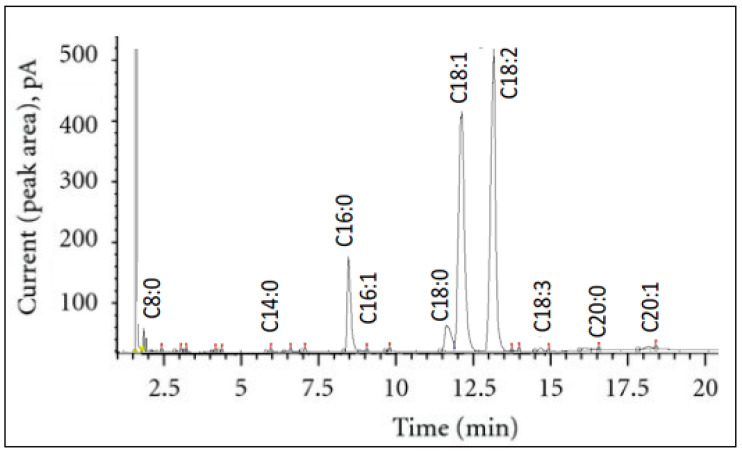
GC-FID chromatogram of fatty acids from sesame seed oil using *n*-hexane solvent for extraction.

**Figure 3 foods-12-03263-f003:**
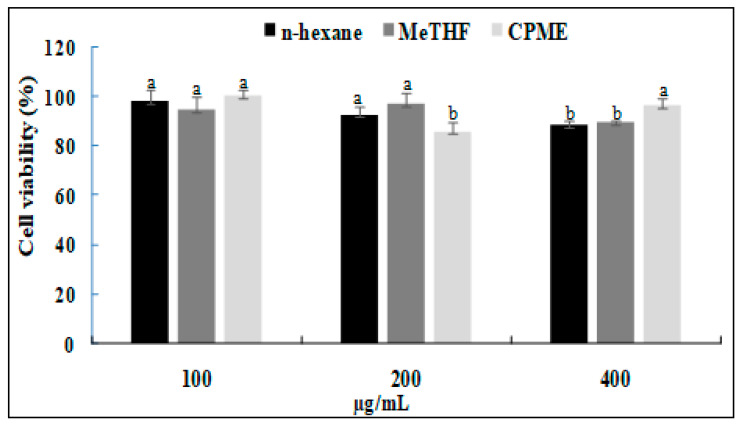
Determination of cell viability after application of sesame seed oils obtained with selected bio-based solvents compared to control. a–b letters = significant differences at *p* < 0.05.

**Figure 4 foods-12-03263-f004:**
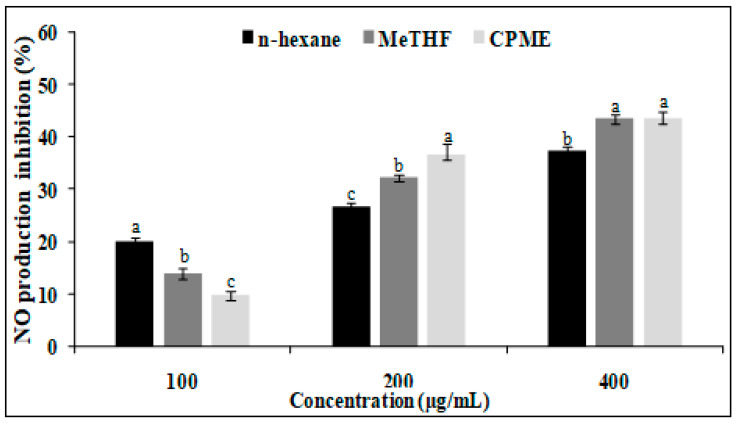
Anti-inflammatory activity of sesame seed oils obtained with selected bio-based solvents compared to control. a–c letters = significant differences at *p* < 0.05.

**Table 1 foods-12-03263-t001:** Solvent properties, including boiling point (°C); Log P. molecular weight (g/mol); viscosity (25 °C); resource; and CMR classification.

		Solvent Properties				
	Bp (°C)	Log P	Mw (g/mol)	Viscosity, (25 °C Cp)	Resource	CMR *
*n*-Hexane	68.5	3.94	86.2	0.31	Petroleum	2
MeTHF	80	0.82	86.1	0.6	Cereal crop	No
CPME	105.3	1.41	100.2	0.55	Chemical synthesis	No
*d*-limonene	175.4	4.45	136.2	0.923	Cereal crop	No
*p*-cymene	173.9	4.02	134.2	0.79	Wood	No
Ethanol	72.6	−0.19	46.1	1.095	Cereal crop	No

BP: Boiling Point; Mw: molecular weight; * CMR classification: Carcinogenic, Mutagenic and/or toxic to Reproduction.

**Table 2 foods-12-03263-t002:** Hansen solubility parameters (δd. δp and δh. In Mpa1/2) and relative energy difference (RED) values for Hansen solubility parameters (HSPs).

	HSPs	RED
	δd	δp	δh	C16:0	C18:1	C18:2	γ-Tocopherol	β-Sitosterol
*n*-Hexane	13.9	0.1	0.1	1.77	1.77	1.95	2.03	2.00
MeTHF	15	4.7	3.9	0.57	0.65	0.75	1.41	1.24
CPME	14.5	3.4	2.8	0.87	0.94	1.17	1.97	1.89
*d*-Limonene	13.9	1.8	2.1	0.73	0.68	0.82	0.31	0.30
*p*-Cymene	14.6	2.2	1.3	1.26	1.12	1.23	1.96	1.87
Ethanol	14	9.1	15.2	2.94	3.03	2.98	5.80	5.90


 Reference or equivalent 
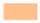
 Better than reference 

 Lesser than reference.

**Table 3 foods-12-03263-t003:** COSMO-RS solubility prediction of major compounds of sesame oil in selected solvents.

	C16:0	C18:1	C18:2	γ-Tocopherol	β-Sitosterol
*n*-Hexane	−1.27	−1.34	−1.31	−0.2974	−0.8017
MeTHF	0.0	0.0	0.0	0.0	0.0
CPME	0.0	0.0	0.0	0.0	0.0
*d*-Limonene	−0.96	−0.98	−0.88	−0.1324	−0.6992
*p*-Cymene	−1.03	−1.03	−0.9	−0.2018	−0.7829
Ethanol	−0.001	−0.05	0.0	−0.8650	−0.5797


 Reference or equivalent 
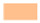
 Better than reference 

 Lesser than reference.

**Table 4 foods-12-03263-t004:** Quality parameters and fatty acid compositions of sesame seed oils extracted with conventional and green solvents.

Quality Parameters	*n*-Hexane	MeTHF	CPME	*d*-Limonene	*p*-Cymene	Ethanol
AV (mgKOH/g oil)	1.69 ± 0.04 ^a^	1.05 ± 0.01 ^a^	1.73 ± 0.02 ^a^	0.92 ± 0.01 ^a^	0.95 ± 0.00 ^a^	1.07 ± 0.04 ^a^
PV (mEq O_2_/kg)	3.42 ± 0.34 ^b^	2.67 ± 0.84 ^b^	3.95 ± 0.37 ^a^	4.83 ± 0.94 ^a^	4.36 ± 0.04 ^a^	4.01 ± 0.02 ^a^
IV (g of I_2_/100 g)	111 ± 2.83 ^a^	123 ± 6.10 ^a^	105 ± 1.89 ^a^	104 ± 0.93 ^a^	93 ± 0.71 ^ab^	96 ± 1.22 ^ab^
RI (20 °C)	1.42 ± 0.04 ^a^	1.45 ± 0.02 ^a^	1.47 ± 0.00 ^a^	1.45 ± 0.01 ^a^	1.44 ± 0.01 ^a^	1.47 ± 0.00 ^a^
K232	0.94 ± 0.00 ^a^	1.33 ± 0.22 ^a^	1.05 ± 0.03 ^a^	1.18 ± 0.05 ^a^	1.52 ± 0.00 ^a^	0.82 ± 0.01 ^a^
K270	0.24 ± 0.01 ^b^	0.38 ± 0.03 ^a^	0.31 ± 0.03 ^a^	0.41 ± 0.00 ^a^	0.47 ± 0.01 ^a^	0.23 ± 0.01 ^b^
Chlorophyll (mg/kg)	2.05 ± 0.45 ^a^	1.89 ± 0.17 ^a^	1.92 ± 0.06 ^a^	2.34 ± 0.04 ^a^	2.67 ± 0.00 ^a^	1.73 ± 0.09 ^a^
Oxidative stability (h)	27.83 ± 0.64 ^a^	28.34 ± 0.25 ^a^	28.56 ± 0.03 ^a^	28.05 ± 0.01 ^a^	27.30 ± 0.66 ^a^	27.04 ± 0.75 ^b^
Fatty acids (%)						
C8:0	0.02 ± 0.01 ^c^	0.43 ± 0.07 ^a^	0.05 ± 0.1 ^b^	0.48 ± 0.1 ^a^	0.45 ± 0.39 ^a^	0.05 ± 0.02 ^c^
C14:0	0.01 ± 0.01 ^c^	0.05 ± 0.02 ^c^	0.08 ± 0.03 ^c^	0.54 ± 0.05 ^a^	0.11 ± 0.02 ^b^	0.01 ± 0.01 ^c^
C16:0	10.71 ± 0.11 ^b^	8.94 ± 0.1 ^b^	8.14 ± 0.01 ^b^	9.57 ± 0.26 ^b^	10.5 ± 0.3 ^a^	9.86 ± 0.17 ^b^
C16:1	0.05 ± 0.03 ^b^	0.32 ± 0.15 ^a^	0.31 ± 0.02 ^a^	0.33 ± 0.04 ^a^	0.33 ± 0.28 ^a^	0.06 ± 0.04 ^b^
C18:0	3.74 ± 0.31 ^c^	5.22 ± 0.04 ^b^	4.15 ± 0.16 ^bc^	5.07 ± 0.64 ^b^	6.67 ± 2.12 ^a^	6.01 ± 0.1 ^a^
C18:1	42.77 ± 0.52 ^a^	39.27 ± 0.27 ^c^	42.08 ± 0.38 ^a^	44.35 ± 2.83 ^a^	40.46 ± 1.51 ^b^	40.79 ± 0 ^b^
C18:2	43.99 ± 0.25 ^a^	42.17 ± 0.26 ^a^	43.41 ± 0.01 ^a^	39.38 ± 1.97 ^b^	38.88 ± 1.65 ^b^	43.18 ± 0.05 ^a^
C18:3	0.33 ± 0.04 ^b^	0.39 ± 0.03 ^b^	0.17 ± 0.1 ^c^	0.14 ± 0.2 ^c^	0.71 ± 0.61 ^a^	0.39 ± 0.01 ^b^
C20:0	0.09 ± 0.02 ^c^	1.07 ± 0.75 ^a^	0.17 ± 0.04 ^bc^	0.10 ± 0.07 ^c^	0.37 ± 0.06 ^b^	0.27 ± 0.13 ^b^
C20:1	0.12 ± 0.03 ^b^	0.23 ± 0.01 ^a^	0.03 ± 0.01 ^c^	0.08 ± 0.02 ^c^	0.20 ± 0.05 ^a^	0.12 ± 0.03 ^b^
∑SFA	14.57 ^c^	15.71 ^b^	12.59 ^b^	15.66 ^b^	18.10 ^a^	16.20 ^b^
∑MUFA	42.94 ^a^	39.82 ^b^	42.42 ^a^	44.76 ^a^	40.99 ^b^	40.97 ^b^
∑PUFA	44.32 ^a^	42.56 ^a^	43.58 ^a^	39.52 ^b^	39.59 ^b^	43.57 ^a^
PUFA/SFA	3.04 ^a^	2.70 ^a^	3.46 ^a^	2.52 ^a^	2.18 ^ab^	2.68 ^a^

Caprylic acid (C8:0); Myristic acid (C14:0); palmitic acid (C16:0); palmitoleic acid (C16:1); stearic acid (C18:0); oleic acid (C18:1); linoleic acid (C18:2); linolenic acid (C18:3); arachidic acid (C20:0); eicosenoic acid (C20:1); a–c letters = significant differences at *p* < 0.05.

**Table 5 foods-12-03263-t005:** Sterol and tocopherol contents in sesame seed oils obtained by bio-based solvents compared to the reference.

	*n*-Hexane	MeTHF	CPME	*d*-Limonene	*p*-Cymene	Ethanol
Sterol contents (mg/100 g oil)						
Cholesterol	1.13 ± 0.01 ^c^	1.47 ± 0.16 ^c^	0.46 ± 0.00 ^d^	4.83 ± 0.02 ^b^	5.15 ± 0.04 ^a^	6.64 ± 0.11 ^a^
Campesterol	99.22 ± 1.72 ^bc^	123.20 ± 1.29 ^b^	164.53 ± 0.83 ^a^	101.00 ± 0.92 ^b^	86.93 ± 0.35 ^c^	66.79 ± 2.64 ^d^
Stigmasterol	33.39 ± 1.86 ^b^	25.89 ± 0.02 ^c^	16.64 ± 0.90 ^d^	29.61 ± 0.73 ^c^	38.11 ± 1.94 ^a^	37.92 ± 1.05 ^a^
*β*-Sitosterol	373.63 ± 2.34 ^c^	437.80 ± 3.94 ^b^	555.93 ± 2.11 ^a^	400.78 ± 1.22 ^c^	364.19 ± 2.03 ^c^	387.96 ± 4.26 ^c^
D-5 Avenasterol	40.21 ± 2.84 ^b^	37.37 ± 1.34 ^bc^	39.48 ± 0.71 ^b^	41.89 ± 0.66 ^b^	44.03 ± 0.03 ^a^	35.22 ± 0.05 ^c^
D-7 Avenasterol	6.13 ± 0.03 ^d^	13.90 ± 0.33 ^a^	8.87 ± 0.21 ^c^	11.83 ± 0.54 ^b^	4.01 ± 0.03 ^e^	5.64 ± 0.04 ^d^
Total	568 ± 8.34 ^c^	641 ± 12.03 ^b^	785 ± 10.92 ^a^	590 ± 23.17 ^c^	543 ± 19.12 ^c^	588 ± 10.08 ^c^
Tocopherol contents (mg/100 g of oil)						
*α*-Tocopherol	1.56 ± 0.13 ^c^	1.61 ± 0.26 ^a^	62.7 ± 0.82 ^d^	1.75 ± 0.91 ^c^	0.83 ± 0.31 ^cd^	1.35 ± 0.73 ^b^
*γ*-Tocopherol	43.44 ± 3.11 ^ab^	46.65 ± 2.45 ^a^	48.74 ± 2.30 ^a^	39.51 ± 1.69 ^b^	39.19 ± 3.04 ^b^	36.37 ± 2.47 ^c^
*δ*-Tocopherol	3.50 ± 0.23 ^b^	2.17 ± 0.04 ^d^	2.92 ± 0.09 ^c^	4.21 ± 0.02 ^a^	3.67 ± 0.02 ^b^	2.06 ± 0.09 ^d^
Total	48 ± 0.56 ^b^	50.6 ± 0.33 ^a^	52.3 ± 1.43 ^a^	44.8 ± 0.92 ^b^	43.7 ± 0.26 ^b^	39.8 ± 0.09 ^c^

a–e letters = significant differences at *p* < 0.05.

**Table 6 foods-12-03263-t006:** Total phenolic contents and antioxidant potential of sesame (*Sesamum indicum* L.) seed oils obtained with *n*-hexane and bio-based solvents.

	TPC (mg GAE/g)	TAC (mg GAE/g)	DPPH Assay (IC_50_ µg/mL)	Reducing Power (EC_50_ µg/mL)
*n*-Hexane	8 ± 0.08 ^c^	1.94 ± 0.72 ^b^	134.64 ± 2.40 ^d^	823.26 ± 4.92 ^d^
MeTHF	22.53 ± 1.18 ^b^	3.14 ± 0.40 ^ab^	90.02 ± 1.39 ^b^	670.15 ± 2.18 ^c^
CPME	23.51 ± 0.09 ^a^	3.92 ± 0.09 ^a^	30.71 ± 0.46 ^a^	435.93 ± 3.84 ^a^
*d*-Limonene	22.35 ± 0.70 ^c^	2.05 ± 0.22 ^b^	101.45 ± 2.45 ^c^	580.62 ± 3.91 ^b^
*p*-Cymene	11.35 ± 0.22 ^e^	1.92 ± 0.54 ^b^	107.23 ± 1.82 ^bc^	1045 ± 2.01 ^e^
Ethanol	21.54 ± 0.82 ^d^	0.34 ± 0.03 ^c^	197.56 ± 2.64 ^e^	807 ± 1.70 ^d^

TPC: Total phenol content; TAC: Total antioxidant potential; GAE: gallic acid equivalent; IC_50_: Half maximal inhibitory concentration; EC_50_: Half maximal effective concentration; a–e letters = significant differences at *p* < 0.05.

## Data Availability

The data presented in this study are available on request from the corresponding author.

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
