# Peer review of "Quality of Edible Sesame Oil as Obtained by Green Solvents: In Silico versus Experimental Screening Approaches"

_foods, 2023, doi:10.3390/foods12173263_

Round 1
Reviewer 1 Report
The manuscript by Trad et al. deals with important topic, nowadays. The use of alternative "green" solvents is of high interest. The authors thoroughly investigated and compared (both theoretically and experimentally) various extraction solvents for sesame seed oil extraction. The manuscript is well written. In my opinion the manuscript is too long and could be at least slightly shortened.
Comments:
1) Table 1: log P value of ethanol is not correct. The unit of viscosity is missing.
2) page 13, "Tocopherol composition", 6th line - alpha tocopherol instead of gamma tocopherol.
3) Table 4: please, add explanation to the superscript letters indicating significance at p level. What is difference among a, b, c and d letters.
4) page 9, statement "Overal, the acid value of strong polar solvent was higher than that of weak polar solvents". This is not clear as n-hexane showed the second highest value (1.69) and n-hexane is not polar.
5) page 15, section 3.6., statement: The results showed....had no cytotoxic effect on RAW cells. In my opinion, there is slight or negligible effect on the cells but not no effect.
6) I would appreciate to see at least one chromatogram in the paper.
Author Response
Response Letter
Dear Doctor,
Please consider the revised version of our article « Quality of Edible Sesame Oil as Obtained by Green Solvents: In Silico versus Experimental Screening Approaches» to be published in « Foods».
The following pages are our point-to-point responses to your comments and suggestions. Besides, the manuscript was thoroughly revised and all errors were corrected. All the corrections were highlighted in yellow in the revised version of manuscript.
The manuscript by Trad et al. deals with important topic, nowadays. The use of alternative "green" solvents is of high interest. The authors thoroughly investigated and compared (both theoretically and experimentally) various extraction solvents for sesame seed oil extraction. The manuscript is well written. In my opinion the manuscript is too long and could be at least slightly shortened.
Comments:
- Table 1: log P value of ethanol is not correct. The unit of viscosity is missing.
Response: log P value of ethanol was corrected( -0.19) and we added the unit of viscosity (Cp)
- page 13, "Tocopherol composition", 6th line - alpha tocopherol instead of gamma tocopherol.
Response: Done, we changed alpha tocopherol instead of gamma tocopherol.
- Table 4: please, add explanation to the superscript letters indicating significance at p What is difference among a, b, c and d letters.
Response: Explanation to the superscript letters indicating significance at p level was added below the table.
- page 9, statement "Overal, the acid value of strong polar solvent was higher than that of weak polar solvents". This is not clear as n-hexane showed the second highest value (1.69) and n-hexane is not polar.
Response: We clarified that ``Meanwhile, increasing the solvent polarity accelerated the destruction of lipids and free fatty acids binding to lipoproteins or cell membranes. Hence, the levels of free fatty acids and fatty alcohols in oils extracted by stronger polar solvents might be higher, resulting in higher acid values. It is also interesting to mention that the acid value is a reflection of pH value of oil; that is as the acid value increases the pH of oil decreases. In fact, the acid value of strong polar solvent was generally higher than that of weak polar solvent but in our study the polar n-hexane solvent showed the second highest value (1.69).``
- page 15, section 3.6., statement: The results showed .... had no cytotoxic effect on RAW cells. In my opinion, there is slight or negligible effect on the cells but not no effect.
Response: We changed «no effect» by «negligible effect» in the revised manuscript.
- I would appreciate to see at least one chromatogram in the paper.
Response: We added a chromatogram of sesame fatty acids in the revised manuscript.
Reviewer 2 Report
Dear authors,
The manuscript provide good information to replace hexane with GRAS solvents on the extraction of sesame oil. For improvement, please consider the following comments:
Introduction: Are there disadvantages on the green solvents evaluated (cost?). If yes, that would be interesting include this information and write a brief sentence informing the reader how these solvents are worthy to be used in terms of product quality and environmental aspects.
Include a topic of reagents and chemicals. Specify brand, country and purity.
Topic 2.3 – Did the authors calculate the yield of oil per extraction? Include the number of replicates used. Include the results as well.
Table 1 – Include the explained acronyms BP and MW in caption.
Tables 2 and 3 – Double check the guide for authors regarding the use of color in table. If not permitted, use symbols, followed with explanations.
Topic 3.2 – That would be great compare the solubility prediction with the experimental parameters related to the yield and quality of oil
Topic 3.4 – Only now I figured out the authors estimated the oil yield. Please, explain how oil yield was estimated in M&M section. How many experimental replicates were done?
Table 5 – standardize the units for both phytosterols and tocopherols.
Topic 3.4 – Antioxidant activity – Replace throughout the text ‘’activity’’ per ‘’potential’’. Remember only the potential of extracts to scavenge free radicals.
Figure 3 - Did the authors mention in M&M the concentrations of extracts tested? At the last sentence of 2.10.2 topic the authors mentioned about NaNO2 standard curve, that sounds confusing. Rewrite the 2.10.2 mentioning how the potential of extract on NO inhibition was calculated. Did the authors use any control or only hexane extract as reference?
Page 13- Tocopherol composition: '' ...(Table 5). γ-tocopherol was the main component followed by δ-tocopherol and γ-tocopherol'' - Fix the mistake
Page 14 - Antioxidant activity: ''Antioxidant activity is an important and necessary criterion to evaluate the quality of vegetable oils'' - What do you mean? Antioxidant potential assays indicates the ability of sample to scavenge free radicals/reduce electrons; if the authors are associating the assay as an association to lipid oxidation that's not accurate.
Page 14 - Antioxidant activity - 3rd paragraph - the authors are comparing your results with other types of assays (ABTS, beta-carotene bleaching); keep consistency on your comparison.
Author Response
Response Letter
Dear Doctor,
Please consider the revised version of our article « Quality of Edible Sesame Oil as Obtained by Green Solvents: In Silico versus Experimental Screening Approaches» to be published in « Foods».
The following pages are our point-to-point responses to your comments and suggestions. Besides, the manuscript was thoroughly revised and all errors were corrected. All the corrections were highlighted in yellow in the revised version of manuscript.
The manuscript provides good information to replace hexane with GRAS solvents on the extraction of sesame oil. For improvement, please consider the following comments:
Introduction: Are there disadvantages on the green solvents evaluated (cost?). If yes, that would be interesting include this information and write a brief sentence informing the reader how these solvents are worthy to be used in terms of product quality and environmental aspects.
Response: In introduction, we added a brief sentence dealing with the disadvantages of green solvents.
Include a topic of reagents and chemicals. Specify brand, country and purity.
Response: We added a topic of reagents and chemicals with brand, country and purity
Topic 2.3 – Did the authors calculate the yield of oil per extraction? Include the number of replicates used. Include the results as well.
Response: Yes, the yield of oil per extraction was calculated and illustrated in the Fig 1 and the result is in the text section 3.4. Experimental solvent screening: Oil yield
Table 1 – Include the explained acronyms BP and MW in caption.
Response: Acronyms BP and MW were added in caption of table 1.
Tables 2 and 3 – Double check the guide for authors regarding the use of color in table. If not permitted, use symbols, followed with explanations.
Response: The guide for authors was checked and there are no restrictions for the use of colors in tables.
Topic 3.2 – That would be great compare the solubility prediction with the experimental parameters related to the yield and quality of oil
Response: In the new text (paragraph: In silico screening versus experimental screening), we added a short pragraph dealing with the comparison of solubility prediction and experimental parameters related to the yield and quality of oil.
Topic 3.4 – Only now I figured out the authors estimated the oil yield. Please, explain how oil yield was estimated in M&M section.
Response: Done in the text (M&M section)
How many experimental replicates were done?
Response: three replicates
Table 5 – standardize the units for both phytosterols and tocopherols.
Response: The units for both phytosterols and tocopherols were standardized
Topic 3.4 – Antioxidant activity – Replace throughout the text ‘’activity’’ per ‘’potential’’. Remember only the potential of extracts to scavenge free radicals.
Response: We replaced throughout the text « activity » by « potential ».
Figure 3 - Did the authors mention in M&M the concentrations of extracts tested?
Response: The concentrations of the tested extracts were added in M&M
At the last sentence of 2.10.2 topic the authors mentioned about NaNO2 standard curve, that sounds confusing. Rewrite the 2.10.2 mentioning how the potential of extract on NO inhibition was calculated. Did the authors use any control or only hexane extract as reference?
Response: Done in the text:
« RAW 264.7 cells were seeded in 24-well plates at a density of 2 105 cells per well and were allowed to attach for 24 h at 37 °C. Then, after 60 min, treated cells with increasing concentrations (100, 200, 400 µg/mL) of extracts (hexane, MeTHF, CPME), dissolved in the DMSO, were stimulated with 100mg/mL lipopolysaccharide (LPS). Cells control were stimulated by LPS without extract treatment.
After 24 h, the quantity of nitrite accumulated in the culture supernatant was determined based on the Griess reaction [34]. The absorbance at 540 nm was then measured and nitric oxide (NO) levels, produced by murine macrophage-like RAW264.7 cells, were determined by comparison with a NaNO2 standard curve. NO production by control and cells treated with sesame oils were compared. »
Page 13- Tocopherol composition: '' ...(Table 5). γ-tocopherol was the main component followed by δ-tocopherol and γ-tocopherol'' - Fix the mistake
Response: The mistake was fixed and corrected in the text. « γ-tocopherol was the main component followed by δ-tocopherol and α –tocopherol ».
Page 14 - Antioxidant activity: ''Antioxidant activity is an important and necessary criterion to evaluate the quality of vegetable oils'' - What do you mean? Antioxidant potential assays indicates the ability of sample to scavenge free radicals/reduce electrons; if the authors are associating the assay as an association to lipid oxidation that's not accurate.
Response: You have reason, we removed this sentence in my manuscript in the result section of antioxidant potential.
Page 14 - Antioxidant activity - 3rd paragraph - the authors are comparing your results with other types of assays (ABTS, beta-carotene bleaching); keep consistency on your comparison.
Response: You have reason, we kept only data which were consistency with our results.
Reviewer 3 Report
The purpose of this study is to evaluate oils obtained from sesame seeds using different solvents and compare them to the traditionally obtained product. As more and more attention is being paid to the environment, the search for new solvents for industrially obtaining oil is of high practical value. The manuscript contains many results, which are described in detail and compared with the literature. However, it requires corrections, a list of which is included below.
Detailed comments
- In the abstract, I recommend to describe more results regarding the nutritional value of the oils obtained, and not to give details of the analytical equipment used.
- The introduction of the manuscript lacks a brief summary of the use of the tested green solvents to extract oil from other plants. Do the results obtained on other plants confirm that the selected solvents give oils with good parameters? The manuscript does not explain why these solvents were chosen for the experiment.
- The article is missing line numbers, so it's hard to point to corrections. These sentences require a stylistic correction. “Although there are numerous studies published about sesame seeds. The investigation of use of green solvents for sesame seed oil extraction are limited”. It should be one sentence?
- The manuscript lacks information about the solvents used. What company did these solvents come from, what purity were those? This is important information, especially when it comes to reagents such as ethanol.
- Minor corrections related to lowercase/uppercase letters: … using An Agilent…., ..three replicates tests were….. Standard of codex Alimentarius
- What standards were used in the chromatographic determination of tocopherols and sterols?
- Table headline number 1 - replace periods with commas
- Why is the presentation of the results in tables not in the order of their discussion? I recommend at least converting the tables number 4, so that the qualitative parameters are included first, and then the fatty acid content.
- The discussion of the results mentions that ... “In Table 4, the differences of oxidative stability, the refractive index, K232, K270 and chlorophyll values of sesame seed oils extracted by n-hexane and five green solvents were not significant (P > 0.05)”….. and in Table 4, the results for K270 differ significantly for n-hexane and ethanol from the other solvents used. Please explain.
- why italicize sesame? Page 10
- …. At the same time, an increase of stearic acid (C18:0) in d-limonene, p-cymene and ethanol oil extracts were observed… Can we talk about an increase or decrease as different solvents were used in the experiment? It should be, a higher or lower value was obtained.
- Why is table number 6 structured differently from the others? I recommend standardizing the tables, and either indicate in all of them what is a traditional solvent and an alternative solvent, or remove these signs from table 6.
- The description of the results of the antioxidant capacity of the oils obtained is too long. There is no need to duplicate the values, which are next to it in the table. Instead, it would be worthwhile to state what is the correlation between antioxidant activity and the content of the studied active components. In this way, it is possible to find out which components have the most effect on the determined activity.
Author Response
Response Letter
Dear Doctor,
Please consider the revised version of our article « Quality of Edible Sesame Oil as Obtained by Green Solvents: In Silico versus Experimental Screening Approaches» to be published in « Foods».
The following pages are our point-to-point responses to your comments and suggestions. Besides, the manuscript was thoroughly revised and all errors were corrected. All the corrections were highlighted in yellow in the revised version of manuscript.
The purpose of this study is to evaluate oils obtained from sesame seeds using different solvents and compare them to the traditionally obtained product. As more and more attention is being paid to the environment, the search for new solvents for industrially obtaining oil is of high practical value. The manuscript contains many results, which are described in detail and compared with the literature. However, it requires corrections, a list of which is included below.
Detailed comments
-In the abstract, I recommend to describe more results regarding the nutritional value of the oils obtained, and not to give details of the analytical equipment used.
Response: We reformulated the abstract.
-The introduction of the manuscript lacks a brief summary of the use of the tested green solvents to extract oil from other plants. Do the results obtained on other plants confirm that the selected solvents give oils with good parameters?
Response: Reviewer 3 suggests that the manuscript is too long and could be at least slightly shortened. So results obtained on other plants that confirm that the selected solvents give oils with good parameters were reported in discussing the results
The manuscript does not explain why these solvents were chosen for the experiment.
Response: We explained why these solvents were chosen for the experiment in introduction and in M&M section (oil extraction).
- The article is missing line numbers, so it's hard to point to corrections. These sentences require a stylistic correction. “Although there are numerous studies published about sesame seeds. The investigation of use of green solvents for sesame seed oil extraction are limited”. It should be one sentence?
Response: Yes, it is one sentence. It was a mistake that has been corrected in the new version.
-The manuscript lacks information about the solvents used. What company did these solvents come from, what purity were those? This is important information, especially when it comes to reagents such as ethanol.
Response: We added a topic of reagents and chemicals with brand, country and purity
-Minor corrections related to lowercase/uppercase letters: … using An Agilent…., ..three replicates tests were….. Standard of codex Alimentarius
Response: These corrections related to lowercase/uppercase letters have been made in the new version of manuscript.
-What standards were used in the chromatographic determination of tocopherols and sterols?
Response: We added the standards used in the chromatographic determination of tocopherols and sterols in M&M section.
-Table headline number 1 - replace periods with commas
Response: The periods were replaced by commas in table headline number 1.
-Why is the presentation of the results in tables not in the order of their discussion? I recommend at least converting the table number 4, so that the qualitative parameters are included first, and then the fatty acid content.
Response: We converted the table 4, the qualitative parameters are included first, and then the fatty acid content.
-The discussion of the results mentions that ... “In Table 4, the differences of oxidative stability, the refractive index, K232, K270 and chlorophyll values of sesame seed oils extracted by n-hexane and five green solvents were not significant (P > 0.05)”….. and in Table 4, the results for K270 differ significantly for n-hexane and ethanol from the other solvents used. Please explain.
Response: You have reason, we checked my results and I found the results for K270 differ significantly for n-hexane and ethanol from the other solvents used. I corrected this in my manuscript:`` In Table 4, the differences of oxidative stability, the refractive index, K232 and chlorophyll values of sesame seed oils extracted by n-hexane and five green solvents were not significant (P> 0.05). For K270, no sig-nificant differences were also observed between solvents with the excep-tion of n-hexane and ethanol.``
- why italicize sesame? Page 10
Response: The italic have been removed. It’s a typo.
- At the same time, an increase of stearic acid (C18:0) in d-limonene, p-cymene and ethanol oil extracts were observed… Can we talk about an increase or decrease as different solvents were used in the experiment? It should be, a higher or lower value was obtained.
Response: In the new version, we replaced increase by a « high proportions ».
-Why is table number 6 structured differently from the others? I recommend standardizing the tables, and either indicate in all of them what is a traditional solvent and an alternative solvent, or remove these signs from table 6.
Response: We removed traditional solvent and an alternative solvent from table 6.
-The description of the results of the antioxidant capacity of the oils obtained is too long. There is no need to duplicate the values, which are next to it in the table. Instead, it would be worthwhile to state what is the correlation between antioxidant activity and the content of the studied active components. In this way, it is possible to find out which components have the most effect on the determined activity.
Response: We reformulated this part and we determinate the correlation between the antioxidant activity and the studied bioactive substances (phenols, sterols and tecopherols):
“As shown in Table 6, the highest total antioxidant potential of sesame seed oils was obtained by CPME (3.92 mg GAE/g) and MeTHF (3.14 mg GAE/g) as compared to the other solvents (1.92-2.05 mg GAE/g). The total antioxidant potential of sesame seed oils was significant correlated with total sterol (r = 0.959), total tocopherol (r = 0.841) and total phenol (r = 0.600) contents. Sesame seed oils extracted using CPME (IC50 = 30.71 μg/mL) and MeTHF (IC50 = 90.02 μg/mL) showed stronger radical scavenging potential than the other green and n-hexane solvents with IC50˃ 100 μg/mL. The antiradical activity of sesame seed oils was significant correlated with total sterol (r = -0.913), total tocopherol (r = 0.700) and total phenol (r = -0.666) contents. Additionally, sesame seed oils extracted using CPME (IC50 = 435.93 μg/mL), d-limonene (IC50 = 580.62 μg/mL) and MeTHF (IC50 = 670.15 μg/mL) showed stronger reducing power potential than the other green and n-hexane solvents with IC50˃ 800 μg/mL). The reducing power activity of sesame seed oils was significant correlated with total sterol (r = -0.882), total phenol (r =- 0.739) and total tocopherol (r = -0.600) contents. So, these results suggested that the antioxidant potential of sesame seed oils could be due to the presence of bioactive substances as phenols, tocopherols and sterols.”
Round 2
Reviewer 3 Report
All my corrections and comments have been included in the revised manuscript